# Characteristics of Individuals with Dry Eye Symptoms without Clinical Diagnosis: Analysis of a Web-Based Survey

**DOI:** 10.3390/jcm8050721

**Published:** 2019-05-21

**Authors:** Ryutaro Yamanishi, Miki Uchino, Motoko Kawashima, Yuichi Uchino, Norihiko Yokoi, Kazuo Tsubota

**Affiliations:** 1Department of Ophthalmology, Keio University School of Medicine, Tokyo 160-8582, Japan; b763_er@a7.keio.jp (R.Y.); motoko-k@a3.keio.jp (M.K.); uchino@z2.keio.jp (Y.U.); tsubota@z3.keio.jp (K.T.); 2Department of Ophthalmology, Kyoto Prefectural University of Medicine, Kyoto 606-8566, Japan; nyokoi@koto.kpu-m.ac.jp

**Keywords:** dry eye disease, undiagnosed dry eye disease, risk factor, self-care

## Abstract

Currently, the available treatment for dry eye disease (DED) varies. The present study aimed to investigate the characteristics of undiagnosed DED and patient-based self-care management for DED based on a web-based survey performed for Tear’s Day in Japan; 1030 participants (301 women) responded; 155 participants (72 women) had a clinical diagnosis of DED. We defined undiagnosed DED (*n* = 116; 54 women) as those with DED symptoms, as evaluated by a frequently used questionnaire despite not having a clinical diagnosis. A multivariate adjusted model indicated that younger age (odds ratio (OR), 0.97 for each one-year decrease; 95% confidence interval (CI), 0.95–0.99), female sex (OR, 2.12; 95% CI, 1.28–3.50), and prolonged visual display terminal usage (OR, 1.12; 95% CI, 1.04–1.21) were risk factors for undiagnosed DED. To investigate the efficacy of self-care management for DED, a sub-analysis was conducted. The number of self-care methods used was significantly higher among women than men. For undiagnosed DED, those with less than three self-care methods had a significantly worse Dry Eye-related Quality-of-Life Score compared with those with diagnosed DED. This study revealed risk factors for undiagnosed DED; individuals with those risk factors need to be clinically assessed and should not rely solely on self-care.

## 1. Introduction

Dry eye disease (DED) is a multifactorial disease of the tears and ocular surface [1]. Symptoms of DED vary and include irritation, dryness, ocular fatigue, foreign body sensation, pain, heavy sensation, brightness, discharge, and fluctuating visual disturbances [2,3,4]; such symptoms can significantly affect quality of life [4,5]. Previous studies have shown that DED is more prevalent in women than in men, and various factors underlying this sex-specific difference have been speculated [5,6,7]. Adverse effects of DED contribute to a substantial economic burden, with indirect costs comprising the largest proportion of the overall cost, due to a substantial loss of work productivity [4]. For instance, DED impairs work performance among office workers, which may lead to a substantial loss to industry [8]. Untreated, DED may become chronic and progressive [5]; therefore, the economic effects of untreated DED should not be ignored.

Moreover, self-care methods for DED have been introduced; however, there have been no reports evaluating the efficacy and differences among the numbers and types of self-care and the symptoms of DED. Individuals attempt to cope with DED symptoms using various methods and tend to find personalized ways [9]. Because many self-care methods are available, to provide more standardized eye treatment and self-care methods, Tear Film-Oriented Therapy (TFOT) has been established in Japan. TFOT for DED treatment is based on the characteristics of the breakup pattern of fluorescein, and targets supplementation for insufficient components of the ocular surface [2,10]. The specific therapy corresponds with unique breakup patterns. To maximize the effect of TFOT, physician visits are essential; accordingly, ocular evaluation by a physician may improve self-care methods.

Many individuals complain about severe DED symptoms, but more than half have not been clinically diagnosed as having DED [11]. As the characteristics of patients with undiagnosed DED (severe symptoms of DED without a clinical diagnosis) and the associated risk factors remain unclear, there is a need to better understand the factors regarding undiagnosed DED. 

In 2017, the Japanese Dry Eye Society enacted 3 July as Namida no Hi (Tear’s Day), with the aim of spreading awareness regarding the importance of tears, to the public at large [12]. In 2018, we performed a web survey to evaluate the demographic characteristics of undiagnosed DED and the relationship with self-care methods for relieving DED symptoms. In particular, this study aimed to investigate the characteristics of and the risk factors for undiagnosed DED, and to compare the findings with patient-based self-care management of DED.

## 2. Materials and Methods 

### 2.1. Study Participants 

We enrolled participants who were willing to complete our web survey; participants were from a registered population of the digital researching company Macromil Incorporated (Tokyo, Japan). Among the 1,200,000 panels, 5000 participants who used a visual display terminal (VDT) during work were randomly selected. We distributed invitation mail to 5000 panels, without introducing the aim of the study. First consecutive 1030 were enrolled in the present study. The survey participants acquired the equivalent of $0.60 for compensation. The information obtained in the survey included age, sex, marital status, offspring status (does or does not have any children), annual household income (<$40,000 per year, $40,000–$59,999 per year, or ≥$60,000), VDT usage hours during work, contact lens (CL) use, smoking history, and symptoms and diagnosis of DED.

This study followed the tenets of the Declaration of Helsinki. The Institutional Review Board of Haneginomori Eye Clinic, Tokyo, Japan, approved the protocol prospectively (ethical approval no. 17009; date: 24 April 2017). Web-based informed consent was obtained from all of the participants after an explanation of the nature and possible consequences of the study.

### 2.2. Dry Eye Disease Symptoms 

We evaluated severe DED symptoms using one of the most frequently used questionnaires for epidemiological studies [13]. The questions used for the evaluation of severe DED symptoms were as follows: (1) “How often do your eyes feel dry (not moist enough)?” and (2) “How often do your eyes feel irritated?”. Possible answers to the two questions concerning symptoms included “constantly”, “often”, “sometimes”, or “never”. Severe DED symptoms were defined as both dryness and irritation, either constantly or often, as previously reported [13].

### 2.3. Diagnosis of Dry Eye Disease

Diagnosed DED was defined as a prior diagnosis of DED. Undiagnosed DED was defined as severe DED symptoms without a physician diagnosis of DED. Non-DED was defined as meeting neither criteria [2].

### 2.4. Dry Eye-Related Quality-Of-Life Score 

The Dry Eye-related Quality-of-Life Score (DEQS) is a questionnaire developed by the Japanese Dry Eye Society [14]. This questionnaire enabled us to evaluate the effect of DED on patients’ daily lives, is often used to evaluate patient-reported symptoms, and is easily used in routine clinical practice [14]. The total score range ranges from 0 to 100, with higher values corresponding to a lower quality of life. The DEQS consists of two components, namely: bothersome ocular symptoms and impact of daily life. The former component includes six contents, as follows: foreign-body sensation, dry sensation in the eyes, painful or sore eyes, ocular fatigue, heavy sensation in the eyelids, and redness in the eyes. The latter component includes nine contents, as follows: difficulty opening eyes, blurred vision when watching something, sensitivity to bright light, problems with eyes when reading, problem with eyes when watching television or looking at a computer or cell phone, feeling distracted because of eye symptoms, eye symptoms affect work, not feeling like going out because of eye symptoms, and feeling depressed because of eye symptoms.

### 2.5. Self-Care Methods for Dry Eye Disease

To study self-care management for DED, the self-care methods that participants used when they felt dry eye symptoms were evaluated. The self-care methods used in the questions were as follows: use of over-the-counter eye drops [15], increase in blinking [16], use protection glasses from dryness [17], use of a warm compress [18], sleeping longer [19], exercise [20], use of oral supplements [15], deep, relaxing breathing [21], and improved eyelid hygiene using eye shampoo [22]. The questions could be answered “yes” or “no”. In addition, the number of self-care methods used was counted (ranging from 0 to 9).

### 2.6. Statistical Analysis

All of the analyses were performed using Statistical Package for Social Sciences (SPSS) Statistics, version 25.0 (IBM Corp., Armonk, NY, USA). Descriptive statistics were used to describe the characteristics of the study population, stratified by sex and subtype of DED diagnosis (non-DED, undiagnosed DED, and diagnosed DED). Using a logistic regression model, we calculated the odds ratios (ORs) and 95% confidence intervals (CIs) of undiagnosed DED and diagnosed DED compared with that of the non-DED group, along with the relevant risks. Firstly, we performed univariate analyses to determine the relationships with each factor. Secondly, we performed multi-adjustment analyses with age, sex, and statistically significant factors identified in univariate analyses. In the second analyses, participants were divided into two groups based on the number of self-care methods used (<3 and ≥3), and were then compared. An unpaired t-test was used for the continuous variables (age, DEQS, and VDT hours), and the chi-squared test was used for the categorical variables. Two-tailed *p* values of <0.05 were considered to indicate a statistically significant difference.

## 3. Results

Among the randomly selected 5000 participants using VDTs during work, the first consecutive 1030 participants took part in the survey. The participant age ranged from 20 to 69 years, and there were 729 men and 301 women. The male participants were significantly older than the female participants (49.6 ± 0.4 years vs. 40.1 ± 0.6 years; *p* < 0.001). The DEQS was higher (worse) in women than in men (17.9 ± 0.7 vs. 24.6 ± 1.2; *p* < 0.001), and the VDT hours during work were longer in women than in men (6.6 ± 0.1 hours vs. 7.1 ± 0.2 hours; *p* = 0.02). Women used CL significantly more frequently compared with the usage among men (16.6% vs. 48.2%; *p* < 0.001). In addition, the number of self-care methods for DED was significantly higher among women than among men (1.6 vs. 1.8; *p* = 0.02; Table 1).

The demographic data for DED diagnosis subgroups are shown in Table 2. One hundred and fifty-five participants (15.0%; 83 men and 72 women) were categorized as having diagnosed DED. One hundred and sixteen (11.3%; 62 men, 54 women) were categorized as having undiagnosed DED (Figure 1). A younger age was more prevalent in the undiagnosed DED group than in the diagnosed DED group (*p* = 0.002). The frequency of female sex was not significant between the undiagnosed DED and diagnosed DED groups (*p* = 0.99). The DEQS was worse in the undiagnosed DED group than in the diagnosed DED group, although the difference was not statistically significant (*p* = 0.06). The DEQS subscale score for bothersome ocular symptoms was significantly higher in the undiagnosed DED group compared with that in the diagnosed DED group (*p* < 0.001). The prevalence of other variables (marital status, offspring status, annual household income, VDT hours during work, CL use, and smoking history) were not significantly different between the groups (Table 2).

A logistic regression model was used to determine the risk factors for the undiagnosed and diagnosed DED. In the multivariate-adjusted model, there was a 3% higher risk for undiagnosed DED for each decrease in one year of age (odds ratio (OR) = 0.97; 95% confidence interval (CI) = 0.95–0.99; *p* = 0.01). In addition, female sex and VDT hours during work were significant risk factors for undiagnosed DED (female sex: OR = 2.16; 95% CI = 1.31–3.33; *p* = 0.003; VDT hours: OR = 1.12; 95% CI = 1.04–1.21; *p* = 0.004). Female sex and CL use were significant risk factors for diagnosed DED (female sex: OR = 2.45; 95% CI = 1.58–3.80; *p* < 0.001; CL use: OR = 1.66; 95% CI = 1.08–2.54; *p* = 0.02; Table 3). 

Table 4 shows the distribution of the self-care methods for DED among the undiagnosed and diagnosed DED groups. The number of self-care methods did not significantly differ between the groups (*p* = 0.37). In the undiagnosed DED group, the most prevalent number of self-care methods was two (31%); in contrast, the most prevalent number in the diagnosed DED group was one (34%). The use of eye drops was the most common self-care method in both groups (undiagnosed, 69.0%; diagnosed, 69.9%; *p* = 0.86). Fourteen participants (12.1%) in the undiagnosed DED group and 15 participants in the diagnosed DED group (9.8%) were not using any self-care methods (*p* = 0.55).

The stratified analysis based on the number of self-care methods (<3 or ≥3) is shown in Figure 2. For the participants using less than three self-care methods, the DEQS was significantly worse in the undiagnosed DED group (*p* = 0.002); however, there was no difference between the undiagnosed and diagnosed DED groups who used three or more self-care methods. 

## 4. Discussion

DED imposes a substantial economic burden, with indirect costs comprising the largest proportion of the overall cost, due to a marked loss of work productivity and a considerable negative impact on physical and potentially psychological function [4]. For instance, Uchino et al. reported that the annual productivity loss owing to DED was estimated to be $6160 per employee when measured by total production, and $1178 per employee when calculated by wage [23]. Based on a previous survey of 4393 Japanese office workers, 26.1% experienced DED symptoms, although only 10% of the workers had a previous clinical diagnosis of DED [24]. Work performance was worse in participants with DED symptoms but without clinical diagnosis of DED compared with those with a clinical diagnosis [8]. Therefore, appropriate treatment and management for such individuals is of paramount importance.

In the present analysis, the multivariate-adjusted model indicated that risk factors for undiagnosed DED included a younger age, female sex, and VDT hours. The multivariate analysis showed a 3% increase in risk of undiagnosed DED for each one year decrease in age; this is in contrast to The Tear Film and Ocular Surface Society Dry Eye Workshop (TFOS DEWS) II report, showing that increasing age is a risk factor for DED, although it is unclear whether the DED was diagnosed or undiagnosed [25]. According to the White Papers and Reports 2009–2010 conducted by the Japanese Ministry of Health, Labor, and Welfare [26], the average income per household as well as per household member by householder age was the highest in the 50–59-year age group ($76,550) and lowest in the ≤29-year age group ($29,890). Moreover, the average income grows in accordance with householder age, until reaching the 50–59-year group. We thus speculate that younger individuals have comparably less money than older individuals, and there is a correspondingly greater impact regarding the cost of visiting the eye clinic, thus younger individuals are less able to afford a visit to an eye clinic as a result of the cost.

In addition, female sex was a risk factor for both undiagnosed DED and diagnosed DED. A previous report shows an increase in DED symptoms in women, even with a similar severity of clinical signs and a lack of association of symptoms with signs [6]. The biological differences between men and women, such as sensitivity of the ocular surface and pain tolerance, and psychological and sociocultural factors, such as self-representation of gender roles, suggest that women tend to be more symptomatic than men [6,7].

Psychological adverse effects, including sleep disorders, are caused by VDT usage [27,28], which may worsen work efficiency and prolonged working time; we therefore also speculate that individuals with a prolonged working time have less time to visit an eye clinic, as most eye clinics are closed at night and in the early morning, except for emergency services. Furthermore, a previous report showed that office workers who spend long hours viewing VDTs develop DED [29], and the blinking rate decreases during using VDTs [30]; thus, workers using VDTs commonly experience eye problems; this is likely more as a result of the longer blinking interval than the decreases in tear breakup time, which disturbs tear stability [31].

Risk factors for diagnosed DED included CL use, although CL use was not a risk factor for undiagnosed DED. The TFOS DEWS II report mentioned that CL use can be one of the main causes of DED [25]. Participants using CL and experiencing severe DED symptoms are considered to have a greater chance of having a DED diagnosis when visiting clinics for the purpose of obtaining prescriptions for CL.

In the present study, women exhibited significantly worse DEQS, despite using more self-care methods compared with that of men. Typically, the relief of DED symptoms requires the use of more self-care methods. Matossian et al. reported that women reported a lower health-related quality of life compared with men [5]; such factors promote the use of a greater number of self-care methods. Moreover, in participants using less than three self-care methods, the DEQS was worse in the undiagnosed DED group than in the diagnosed DED group; we speculate that those with undiagnosed DED may not perform self-care management appropriately. The concept of TFOT is layer-by-layer treatment for DED, with each therapy corresponding to its target layer; this is proposed so as to assist with the supplementation of the corneal surface components that are insufficient [10]. Self-care, such as a warm compress and eyelid hygiene, are recommended for the lipid layer, and artificial tears are recommended for the aqueous layer [10]. In addition, the specific therapy corresponds with breakup patterns, as follows: area break or line break, punctal plug and diquafosol sodium; spot break, dimple break, and line or random break with rapid expansion, diquafosol sodium and rebamipide; and random break, Meibomian Gland Dysfunction therapy, ointment, artificial tear, hyaluronic acid, diquafosol sodium, and rebamipide [10]. Without an appropriate understanding of and information concerning TFOT, patient-based self-care may not be properly initiated. Thus, it is possible that those experiencing severe DED symptoms are more likely to visit an eye clinic and obtain information for the appropriate management of DED.

Currently, there are various available treatment options for DED; however, even for eye professionals, the widely diverse management options can be confusing [32,33]. The TFOS DEWS II report summarized and recommended appropriate management and treatment for DED based on an evidence-based review of the literature, and it encouraged education regarding DED and its management, treatment, and prognosis [15] However, although eyelid hygiene and warm compresses are effective self-care methods for Meibomian gland disease, compliance has not been well characterized [15,18,34]. To maximize the effectiveness of DED treatment, self-care management should be provided based on ocular status. Accordingly, it should be recommended that patients should visit an eye care physician to obtain appropriate information.

We acknowledge several limitations to the present study. Firstly, it relied on self-reporting to assess the presence of DED as either clinically diagnosed or as undiagnosed based on symptoms; no objective examination for DED was performed. Secondly, the web survey answers were recorded consecutively; therefore, the responses were disproportionate by age and sex. There may also have been a selection bias in that individuals who are interested in eye symptoms may have been more likely to participate. There is a need for a study wherein age and sex are appropriately distributed. Lastly, this is a cross-sectional study, thus any potential causal relationship remains unclear. For example, there may be reverse causality among a lower DEQS and the number of self-care methods. In addition, participants with severe symptoms may have had a lower DEQS owing to challenges implementing self-care or inadequately employing self-care.

In conclusion, the present study revealed that a younger age, female sex, and VDT hours during work are risk factors for undiagnosed DED. Furthermore, those in the undiagnosed DED group and those using less than three self-care methods may perform self-care inappropriately, and thus require professional support. Although further studies are needed to increase the understanding of the importance of appropriate management for DED, the present results suggest the need to increase the awareness of both undiagnosed DED and appropriate self-care for DED. 

## Figures and Tables

**Figure 1 jcm-08-00721-f001:**
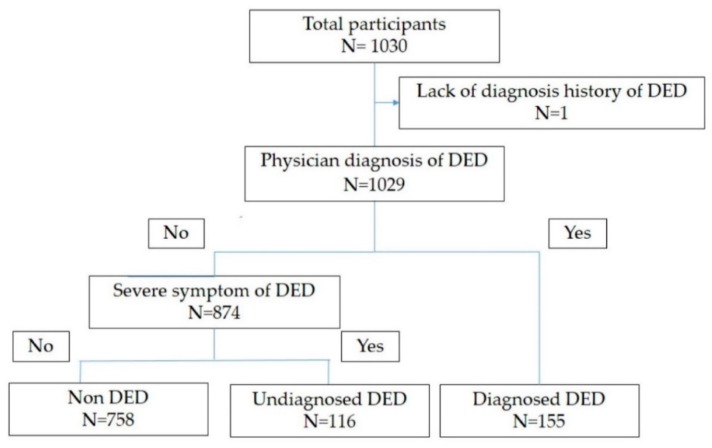
Participants flow chart. The participants were stratified with diagnosed history and symptom of dry eye disease (DED). Diagnosed DED—physician diagnosis of DED (+); undiagnosed DED—physician diagnosis of DED (−) and severe symptoms of DED (+); non DED—physician diagnosis of DED (−) and severe symptoms of DED (−).

**Figure 2 jcm-08-00721-f002:**
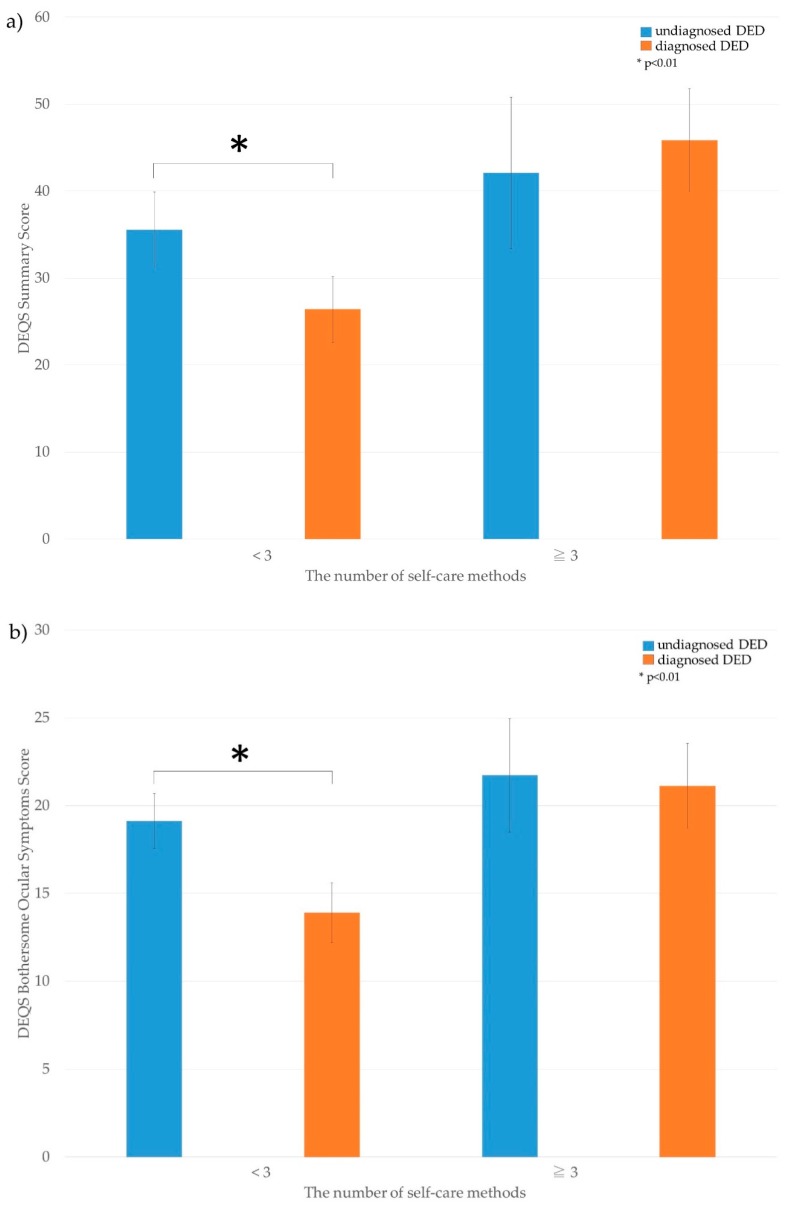
The number of self-care methods and Dry Eye-related Quality of Life Score among undiagnosed and diagnosed dry eye disease. The difference in Dry Eye-related Quality of Life Score (DEQS) between undiagnosed/diagnosed dry eye disease (DED), stratified with the amount of self-care items for DED. (**a**) DEQS summary score; (**b**) DEQS bothersome ocular symptoms score; (**c**) DEQS impact of daily life score.

**Table 1 jcm-08-00721-t001:** The distribution of the study population.

Demographic Characteristic	Men	Women	*p* Value
	*n* (%)	*n* (%)	
	729 (70.8)	301 (29.2)	
Age (year), mean ± SD	49.6 ± 0.4	40.1 ± 0.6	<0.001 ^a^
DEQS			
Summary score, mean ± SD	17.9 ± 0.7	24.6 ± 1.2	<0.001 ^a^
Bothersome ocular symptoms, mean ± SD	9.0 ± 0.3	13.1 ± 0.5	0.002 ^a^
Impact of daily life, mean ± SD	8.9 ± 0.4	11.4 ± 0.7	<0.001 ^a^
Marital status: married *n* (%)	502 (68.9)	111 (36.9)	<0.001 ^b^
Offspring status: does have any children *n* (%)	436 (59.8)	86 (28.6)	<0.001 ^b^
Annual household income *n* (%)			<0.001^b^
<$40,000 per year	94 (12.9)	87 (28.9)	
$40,000–$59,999 per year	145 (19.9)	51 (16.9)	
≥$60,000 per year	430 (59.0)	116 (38.5)	
VDT hours during work (hrs), mean ± SD	6.6 ± 0.1	7.1 ± 0.2	0.02 ^a^
CL use, *n* (%)	121 (16.6)	145 (48.2)	<0.001 ^b^
History of smoking, *n* (%)	202 (27.7)	47 (15.6)	<0.001 ^b^
The number of self-care methods mean + SD	1.6 (0.0)	1.8 (0.1)	0.02 ^a^

DEQS—Dry Eye-related Quality of Life Score; VDT—visual display terminal; CL—contact lens; SD—standard deviation. ^a^ an unpaired two-tailed test; ^b^ the chi-square test.

**Table 2 jcm-08-00721-t002:** The distribution of the study population among dry eye categories.

Demographic Characteristic	Non DED	Undiagnosed DED	Diagnosed DED	*p* Value **
	*n* (%)	*n* (%)	*n* (%)	
	758 (73.7)	116 (11.3)	155 (15.1)	
Age (year), mean ± SD	47.8 ± 0.4	41.8 ± 0.9	45.7 ± 0.9	0.002 ^a^
Women, *n* (%)	175 (23.1)	54 (46.6)	72 (46.5)	0.99 ^b^
DEQS				
Summary score, mean ± SD	14.6 ± 0.5	37.3 ± 2.0	32.2 ± 1.8	0.06 ^a^
Bothersome ocular symptoms, mean ± SD	7.6 ± 0.2	19.8 ± 0.7	16.0 ± 0.7	<0.001 ^a^
Impact of daily life, mean ± SD	7.0 ± 0.3	17.5 ± 1.4	16.2 ± 1.1	0.47 ^a^
Marital status: married, *n* (%)	463 (61.1)	62 (53.4)	88 (56.8)	0.59 ^b^
Offspring status: no children, *n* (%)	400 (52.8)	55 (47.4)	67 (43.2)	0.49 ^b^
Annual household income, *n* (%)				0.82 ^b^
<$40,000/year	127 (16.8)	26 (22.4)	28 (18.1)	
$40,000–$59,999/year	143 (18.9)	23 (55.0)	30 (19.4)	
≥$60,000/year	414 (54.6)	57 (49.1)	75 (48.4)	
VDT hours during work (hours), mean ± SD	6.6 ± 0.1	7.7 ± 0.2	7.2 ± 0.2	0.08 ^a^
CL use, *n* (%)	164 (21.6)	41 (35.3)	61 (39.4)	0.50 ^b^
History of smoking, *n* (%)	186 (24.5)	22 (19.0)	41 (26.5)	0.14 ^b^

DED—dry eye disease; DEQS—Dry Eye-related Quality of Life Score; VDT—visual display terminal; CL—contact lens; SD—standard deviation. ** analysis between undiagnosed DED and diagnosed DED. ^a^ an unpaired two-tailed test. ^b^ the chi square test.

**Table 3 jcm-08-00721-t003:** Risk factors of undiagnosed and diagnosed dry eye disease.

Demographic Characteristic	Undiagnosed DED			Diagnosed DED		
	Univariate Logistic Analysis (Compared to Non-DED)	Multivariate Logistic Analysis (Compared to Non-DED)	*p* Value	Univariate Logistic Analysis (Compared to Non-DED)	Multivariate Logistic Analysis (Compared to Non-DED)	*p* Value
Age (every one year)						
OR (95% CI)	0.95 (0.93–0.97)	0.97 (0.95–0.99)	0.01	0.98 (0.97–1.00)	1.01 (0.99–1.03)	0.38
Gender (men vs. women)						
OR (95% CI)	2.91 (1.94–4.34)	2.12 (1.28–3.50)	0.003	2.89 (2.02–4.14)	2.45 (1.58–3.80)	<0.001
Offspring status: does have any children						
OR (95% CI)	0.81 (0.55–1.20)			0.68 (0.48–0.97)	0.83 (0.55–1.26)	0.39
Annual household income (every $20,000)						
OR (95% CI)	0.90 (0.81–0.99)	0.96 (0.87–1.05)	0.38	1.00 (0.92–1.08)		
VDT hours during work (every one hour)						
OR (95% CI)	1.15 (1.07–1.23)	1.12 (1.04–1.21)	0.004	1.08 (1.01–1.15)	1.06 (0.99–1.14)	0.07
CL use (never vs. ever)						
OR (95% CI)	2.01 (1.29–3.12)	1.28 (0.78–2.11)	0.33	2.35 (1.63–3.39)	1.66 (1.08–2.54)	0.02

DED—dry eye disease; OR—odd ratio; CI—confidence interval; CL—contact lens; VDT—visual display terminal.

**Table 4 jcm-08-00721-t004:** The distribution of self-care methods among undiagnosed and diagnosed dry eye disease.

Self-Care Methods	Undiagnosed DED	Diagnosed DED	*p* Value
The number of methods, mean + SD	1.9 ± 0.1	2.0 ± 0.1	0.37 ^a^
The number of methods, *n* (%)			0.50 ^b^
0	14 (12.1)	15 (9.8)	
1	35 (30.2)	52 (34.0)	
2	36 (31.0)	37 (24.2)	
≥3	31 (26.7)	49 (32.0)	
Use of over-the-counter eye drops, *n* (%)	80 (69.0)	107 (69.9)	0.86 ^b^
Increase in blinking	35 (30.2)	38 (24.8)	0.33 ^b^
Use protection glasses from dryness	6 (5.2)	12 (7.8)	0.39 ^b^
Use of a warm compress	24 (20.7)	36 (23.5)	0.58 ^b^
Sleeping longer	42 (36.2)	71 (46.4)	0.09 ^b^
Exercise	4 (3.4)	12 (7.8)	0.13 ^b^
Use of oral supplements	13 (11.2)	13 (8.5)	0.46 ^b^
Deep, relaxing breathing	7 (6.0)	13 (8.5)	0.45 ^b^
Eyelid hygiene using eye shampoo	2 (1.7)	0 (0.0)	0.10 ^b^
Others	3 (2.6)	6 (3.9)	0.55 ^b^
Not doing anything	14 (12.1)	15 (9.8)	0.55 ^b^

DED—dry eye disease; SD—standard deviation; ^a^ an unpaired two-tailed test; ^b^ the chi-square test.

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
