# Peer review of "Characteristics of Individuals with Dry Eye Symptoms without Clinical Diagnosis: Analysis of a Web-Based Survey"

_jcm, 2019, doi:10.3390/jcm8050721_

Round 1
Reviewer 1 Report
Interesting, highly detailed report regarding undiagnosed dry eye disease in Japan.
Exploring the demographics and symptoms of patients who have not
received an actual clinical diagnosis is not typically performed for
disease and is highly useful. It captures data on a subgroup of
patients outside the awareness of doctors- this approach deserves greater use in
other diseases. The collection of data and analysis was excellent and
resulted in strong conclusions. It is also helpful for eye care
specialists to understand that a Tear Film-Oriented Therapy (TFOT) is standard
of care in Japan. Although DED is a disease primarily of interest to
ophthalmologists, other specialties examining patients may now be more
likely to ask about symptoms of DED. Publication of the paper may also
encourage further exploration of non-diagnosed disease in other
specialties.
Suggestion: Because of potential interest in TFOT , some additional explanation of this would be helpful.
Author Response
Title: Characteristics of individuals with dry eye symptoms without clinical diagnosis: analysis of a web-based survey
Point by Point
Author's Reply to the Review Report (Reviewer 1)
Comment:
Interesting, highly detailed report regarding undiagnosed dry eye disease in Japan. Exploring the demographics and symptoms of patients who have not received an actual clinical diagnosis is not typically performed for disease and is highly useful. It captures data on a subgroup of patients outside the awareness of doctors- this approach deserves greater use in other diseases. The collection of data and analysis was excellent and resulted in strong conclusions. It is also helpful for eye care specialists to understand that a Tear Film-Oriented Therapy (TFOT) is standard of care in Japan. Although DED is a disease primarily of interest to ophthalmologists, other specialties examining patients may now be more likely to ask about symptoms of DED. Publication of the paper may also encourage further exploration of non-diagnosed disease in other specialties.
Reply:
Thank you for understanding our study, its intent, and for your positive comment.
Comment:
Suggestion: Because of potential interest in TFOT, some additional explanation of this would be helpful.
Reply:
Thank you for bringing this important issue to our attention. We agree with your suggestion and added some sentences in the discussion section as follows ‘’.
The treatment for DED depend on the break up pattern of fluorescein, and classified as three types. Firstly, in case area break or line break are observed, recommended treatment are punctal plug and diquafosol sodium. Secondly, diquafosol sodium and rebamipide are recommended in patient whose break up pattern is spot break, dimple break, and line or random break with rapid expansion. Thirdly, patient with random break pattern should be recommended Meibomian Gland Dysfunction (MGD) care, ointment, artificial tear, hyaluronic acid, diquafosol sodium and rebamipide. Following strategies are introduced in the article (Yokoi, N.; Georgiev, G.A. Tear Film-Oriented Diagnosis and Tear Film-Oriented Therapy for Dry Eye Based on Tear Film Dynamics. Invest. Ophthalmol. Vis. Sci. 2018, 59, Des13-des22, doi:10.1167/iovs.17-23700.).
Change in the Manuscript:
We added highlighted sentences in 4. Discussion section as follows ‘Self-care, such as a warm compress and eyelid hygiene, are recommended for the lipid layer, and artificial tears are recommended for the aqueous layer [10]. In addition, the specific therapy corresponds with breakup pattern of fluorescein as follows: area break or line break, punctal plug and diquafosol sodium; spot break, dimple break, and line or random break with rapid expansion, diquafosol sodium and rebamipide; random break, Meibomian Gland Dysfunction therapy, ointment, artificial tear, hyaluronic acid, diquafosol sodium and rebamipide [10].’.
Reviewer 2 Report
This is quite interesting study led by Yamanishi, R. In my opinion, some parts of the manuscript can be improved:
(1) How the web survey was conducted? Was it through a blog or through direct email? If through direct email, then how the information about the subjects were extracted?
(2) Properly define DEQS questionnaire and self-care methods. Why the authors did not use the SPEED questionnaire for the evaluation of the dry eye disease?
(3) In 2017, the dry eye society reacted to have announced the Tear's day. Why the symptoms of dry eye are increased in Japan? Is it related to the pollution of certain parts of Japan? Why the study only selected the participants using VDTs during work? Is there any prior study that shows that the subjects using VDTs during work have got more chance of developing DED?
(4) Can the authors properly define undiagnosed-DED and non-DED?
Author Response
Title: Characteristics of individuals with dry eye symptoms without clinical diagnosis: analysis of a web-based survey
Point by Point
Comment:
This is quite interesting study led by Yamanishi, R. In my opinion, some parts of the manuscript can be improved.
Reply:
We appreciate your dedication to improve our manuscript and your positive comment.
Comment:
(1) How the web survey was conducted? Was it through a blog or through direct email? If through direct email, then how the information about the subjects were extracted?
Reply:
Thank you for your valuable comment. This survey was performed through neither blog nor direct email. We announced this web survey to 1,200,000 subjects who are registered as candidates for the survey of Macromil Incorporated (digital researching company). Then 5,000 subjects who used a VDT during work were randomly selected. First 1030 participants were recorded consecutively.
Change in the Manuscript:
We added highlighted sentences in 2.1. Study participants section as follows ‘We enrolled participants who were willing to complete our web survey; participants were from a registered population of the digital researching company Macromil Incorporated (Tokyo, Japan). Among 1,200,000 panels, 5,000 participants who used a visual display terminal (VDT) during work were randomly selected. We distributed invitation mail to 5,000 panels without introducing the aim of the study. First consecutive 1030 were enrolled in this study.’.
Comment:
(2) Properly define DEQS questionnaire and self-care methods. Why the authors did not use the SPEED questionnaire for the evaluation of the dry eye disease?
Reply:
We appreciate your meaningful comment. DEQS is validated questionnaire for evaluating effect of DED symptom on individuals’ quality of life (Sakane, Y.; Yamaguchi, M.; Yokoi, N.; Uchino, M.; Dogru, M.; Oishi, T.; Ohashi, Y.; Ohashi, Y. Development and validation of the Dry Eye-Related Quality-of-Life Score questionnaire. JAMA ophthalmology 2013, 131, 1331-1338,). Please note that the detail of DEQS is stated in 2.4. Dry Eye-Related Quality-of-Life Score section, line 61 to 76.
Although we acknowledge the importance of proper definition of self-care methods in DED, there was no previous report summarizing all of them. Our study is the first one which refers to the self-care methods.
For the SPEED questionnaire, we acknowledge it can to quickly track the progression of dry eye symptoms over time. Unfortunately, the SPEED questionnaire has not been validated in Japanese. Therefore, we could not use in this study.
Comment:
(3) In 2017, the dry eye society reacted to have announced the Tear's day. Why the symptoms of dry eye are increased in Japan? Is it related to the pollution of certain parts of Japan? Why the study only selected the participants using VDTs during work? Is there any prior study that shows that the subjects using VDTs during work have got more chance of developing DED?
Reply:
We thank you for bringing this to our attention.
In the recent study, more than 20 million Japanese people were considered as having DED symptom, and the amount has been increasing rapidly. The popularization of air conditioning system, increasing VDT user and CL wearer, and becoming aging society account for the reason (Yokoi, N.; Kato, H., The Paradigm Shift of Dry Eye in the Clinical Setting; Tear Film Oriented Diagnosis and Therapy. Journal of Kyoto Prefectural University of Medicine 2013, 122, 549-558. article in Japanese.)
Although air pollution is associated with high risk of DED in Taiwan (Zhong, J.Y.; Lee, Y.C.; Hsieh, C.J.; Tseng, C.C.; Yiin, L.M. Association between Dry Eye Disease, Air Pollution and Weather Changes in Taiwan. Int. J. Environ. Res. Public Health 2018, 15, doi:10.3390/ijerph15102269.), there was no report regarding the association between pollution and DED in Japan.
Uchino et al. previously reported the prevalence of DED among Japanese office workers who use VDT and revealed that office workers who spend long hours viewing VDT develop DED (Uchino, M.; Yokoi, N.; Uchino, Y.; Dogru, M.; Kawashima, M.; Komuro, A.; Sonomura, Y.; Kato, H.; Kinoshita, S.; Schaumberg, D.A., et al. Prevalence of dry eye disease and its risk factors in visual display terminal users.